# Monocyte subset redistribution from blood to kidneys in patients with Puumala virus caused hemorrhagic fever with renal syndrome

Sindhu Vangeti[1], Tomas Strandin[2], Sang Liu[1], Johanna Tauriainen[3], Anne Räisänen-Sokolowski[4], Luz Cabrera[2], Antti Hassinen[5], Satu Mäkelä[6], Jukka Mustonen[6,7], Antti Vaheri[2], Olli Vapalahti[2], Jonas Klingström[3], Anna Smed-Sörensen[1] *

1 Division of Immunology and Allergy, Department of Medicine Solna, Karolinska Institutet, Karolinska University Hospital Stockholm, Sweden, 2 Department of Virology, Faculty of Medicine, University of Helsinki, Helsinki, Finland, 3 Center for Infectious Medicine, Department of Medicine Huddinge, Karolinska Institutet, Stockholm, Sweden, 4 Department of Pathology, Helsinki University Hospital and Helsinki University, Helsinki, Finland, 5 Institute for Molecular Medicine Finland (FIMM), University of Helsinki, Helsinki, Finland, 6 Department of Internal Medicine, Tampere University Hospital, Tampere, Finland, 7 Faculty of Medicine and Health Technology, Tampere University, Tampere, Finland

she These authors contributed equally to this work.
* anna.smed.sorensen@ki.se

**Data Availability Statement:** All relevant data are within the manuscript and its Supporting Information files.

## Abstract

Innate immune cells like monocytes patrol the vasculature and mucosal surfaces, recognize pathogens, rapidly redistribute to affected tissues and cause inflammation by secretion of cytokines. We previously showed that monocytes are reduced in blood but accumulate in the airways of patients with Puumala virus (PUUV) caused hemorrhagic fever with renal syndrome (HFRS). However, the dynamics of monocyte infiltration to the kidneys during HFRS, and its impact on disease severity are currently unknown. Here, we examined longitudinal peripheral blood samples and renal biopsies from HFRS patients and performed in vitro experiments to investigate the fate of monocytes during HFRS. During the early stages of HFRS, circulating CD14–CD16+ nonclassical monocytes (NCMs) that patrol the vasculature were reduced in most patients. Instead, CD14+CD16– classical (CMs) and CD14+CD16+ intermediate monocytes (IMs) were increased in blood, in particular in HFRS patients with more severe disease. Blood monocytes from patients with acute HFRS expressed higher levels of HLA-DR, the endothelial adhesion marker CD62L and the chemokine receptors CCR7 and CCR2, as compared to convalescence, suggesting monocyte activation and migration to peripheral tissues during acute HFRS. Supporting this hypothesis, increased numbers of HLA-DR+, CD14+, CD16+ and CD68+ cells were observed in the renal tissues of acute HFRS patients compared to controls. In vitro, blood CD16+ monocytes upregulated CD62L after direct exposure to PUUV whereas CD16– monocytes upregulated CCR7 after contact with PUUV-infected endothelial cells, suggesting differential mechanisms of activation and response between monocyte subsets. Together, our findings suggest that NCMs are reduced in blood, potentially via CD62L-mediated attachment to endothelial cells and monocytes are recruited to the kidneys during HFRS. Monocyte

**Funding:** This work was supported by grants to AS-S from the Swedish Research Council (2012-02088, 2017-01026), the Swedish Heart-Lung Foundation (20140591, 20160246, 20180073, 20190170), the Swedish Childhood Cancer Fund (TJ2014-0030) and Karolinska Institutet (2-3591/2014). SV was supported by grants from the Swedish Heart-Lung Foundation (20180817) and Karolinska Institutet (2018-02663, 2018-02831, 2019-00228, 2019-00320, 2019-01419). TS was supported by grants from Academy of Finland (321809) and University of Helsinki. JK was supported by a grant from Swedish Research Council (2018-02646). JM was supported by the Competitive State Research Funding of the Expert Responsibility Area of Tampere University Hospital (9X033, 9AA050), the Tampere Tuberculosis Foundation and the the Sigrid Jusélius Foundation. AV was supported by grants from the Sigrid Jusélius Foundation and the Magnus Ehrnrooth Foundation. OV was supported by a grant from the Sigrid Jusélius Foundation. The funders had no role in study design, data collection and analysis, decision to publish, or preparation of the manuscript.

**Competing interests:** The authors have declared that no competing interests exist.

mobilization, activation and functional impairment together may influence the severity of disease in acute PUUV-HFRS.

## Author summary

Hantaviruses are re-emerging human pathogens that can cause severe disease, typically manifesting in the lungs or kidneys. The virus preferentially infects the endothelial cells without killing them. Therefore, the vascular leakage associated with hantavirus disease, and hemorrhagic fever with renal syndrome (HFRS) is believed to be a consequence of the dysregulated immune response to infection. In the present study, in a cohort of PUUV-infected patients with acute HFRS, we describe a striking depletion of nonclassical monocytes from circulation while classical and intermediate monocyte frequencies are increased. Importantly, we found increased numbers of cells expressing monocyte and macrophage markers in the kidneys of patients with HFRS. The monocytes remaining in circulation show signs of activation, migration to the periphery and impairment in their ability to respond to TLR stimulation. Interestingly, the magnitude of monocyte activation was associated with greater disease severity. In addition, we also noted that different monocyte subsets differ in how they recognize and respond to cell-free or cell-associated hantavirus exposure. Collectively, our study greatly adds to what is currently known about how monocytes behave during human hantavirus disease, and highlights the importance of studying functional differences across the major monocyte subsets at greater resolution.

## Introduction

Hantaviruses (genus *Orthohantavirus*, family *Hantaviridae*, Order *Bunyavirales*) are zoonotic viruses that pose a clear burden to human health [1,2]. Hantaviruses enter the body primarily via inhalation and infection is typically accompanied by viremia, which suggests that circulation is a likely route of systemic dissemination for the virus [3]. The severity and symptoms of the hantavirus-caused human diseases depend on the causative virus species. Hantaviruses of Eurasia (i.e. "old world") cause hemorrhagic fever with renal syndrome (HFRS) whereas hantaviruses of the Americas (i.e. "new world") can cause hantavirus pulmonary syndrome (HPS) [1,4–6]. Although the most affected organ varies from kidneys in HFRS to lungs in HPS, a hallmark of both diseases is increased vascular leakage. Microvascular endothelial cells, that form the barrier between the capillary bloodstream and tissues, are the prime target cells of hantavirus infection in humans but the virus itself does not directly affect endothelial cell permeability. Instead, vascular leakage is likely a consequence of an excessive immune response to hantavirus infections [6–8], and several immune-related processes have been suggested to be involved [9–11].

Monocytes are critical mediators of innate immune responses at mucosal surfaces and in blood. In addition, monocytes can migrate to peripheral tissues. Monocytes interact closely with endothelial cells during extravasation [12], which further increases the importance of investigating monocytes in the context of hantavirus infection, known to affect endothelial cell barrier functions. Three distinct monocyte subsets have been described based on their surface expression of CD14 and CD16: CD14+CD16– classical monocytes (CM), CD14+CD16+ intermediate monocytes (IM) and CD14–CD16+ nonclassical monocytes (NCM) [13]. CMs can translocate from blood through the endothelium into tissues, and under inflammatory conditions take on macrophage- or dendritic cells (DC)-like properties [13]. IMs are potent cytokine

producers and expand in response to infections, sepsis and vaccinations [13,14]. NCMs, also called patrolling monocytes since they "crawl" on the endothelium [15], are well-equipped to sense danger signals such as virus infections [16].

Monocyte transmigration through the endothelium is dependent on interactions with adhesion proteins, many of which are shared between several leukocytes and endothelial cells [17]. During inflammatory conditions monocytes are captured from free-flowing blood by endothelial P- and E-selectins or through monocyte L-selectin (CD62L) and endothelial CD34 interactions [18]. The initial capture is followed by firm monocyte arrest mediated through interactions between monocyte integrins and EC-expressed intercellular adhesion molecule 1 (ICAM1) and vascular cell adhesion molecule 1 (VCAM1). Elevated levels of ICAM1 are detected in HFRS patients [19,20] and in hantavirus-infected endothelial cells in vitro [9,21], indicating that the virus-infected endothelium does have strong propensity to attract monocytes during acute HFRS. The migration of monocytes to the site of infection is mediated by soluble chemokines that interact with chemokine receptors expressed on the cell surface. Monocytes egress from the bone marrow and infiltration into tissues is dependent on CCL2 and CCL3 interactions with CCR2 [22]. Monocytes can also migrate efficiently through CCR6, similar to observations in macrophages in various disease models [23,24]. CCR7, on the other hand, guides trafficking to the lymph nodes [25,26].

Here we describe an early reduction of circulating NCMs during severe acute HFRS. Concurrent increased hematopoietic progenitor frequencies in blood suggest their mobilization from the bone marrow to replenish the reduced circulating NCM pool. Increased numbers of HLA-DR+, CD14+, CD16+ and CD68+ cells in kidney tissues of acute HFRS indicated pronounced monocyte extravasation from blood into tissues during HFRS. Monocyte kinetics during HFRS, and expression of activation, adhesion and migration markers together suggest a direct role for monocytes in kidney pathology during HFRS.

## Results

### Study design and patient characteristics

Longitudinal blood samples were obtained from 23 patients hospitalized due to PUUV-caused HFRS during acute (5–10 days after symptom onset) and convalescence stages (11–30, 180 and 360 days after onset of symptoms) of the disease (Fig 1A and Table 1). Blood samples from 9 age- and sex-matched uninfected controls (UC) were also included. Thrombocyte counts, creatinine values and mean arterial pressure (MAP) values were used to calculate adapted sequential organ failure assessment (a-SOFA) scores, to stratify patients by HFRS disease severity (Table 2). Thirteen out of 23 (57%) patients were current smokers. Severe thrombocytopenia ($< 50$ x $10^9$/L) was found in 3 patients, severe acute kidney injury (AKI) (defined as the highest plasma creatinine concentration $>353.6$ μmol/L [27]) in 8 patients. Three patients were hypotensive with lowest measured MAP $< 70$ mmHg. Characteristic for HFRS, the patients had significantly increased plasma levels of creatinine (Fig 1B) and C-reactive protein (CRP) (Fig 1C) as well as decreased levels of blood thrombocytes (Fig 1D) during acute stage. As expected [28], a majority (13 out of 17) patients with plasma samples available had detectable viremia during the acute phase of HFRS (Fig 1E).

### Patients with more severe HFRS display delayed peak monocyte frequencies

We found that patients with severe disease as defined by a-SOFA score $\geq 5$ had slightly elevated total blood leukocyte counts compared to patients with mild disease (a-SOFA score $<5$) (Fig 1F). Furthermore, the frequency of total monocytes was elevated during acute HFRS as compared to UC but more pronounced in patients with a-SOFA scores $\geq 5$ than in patients

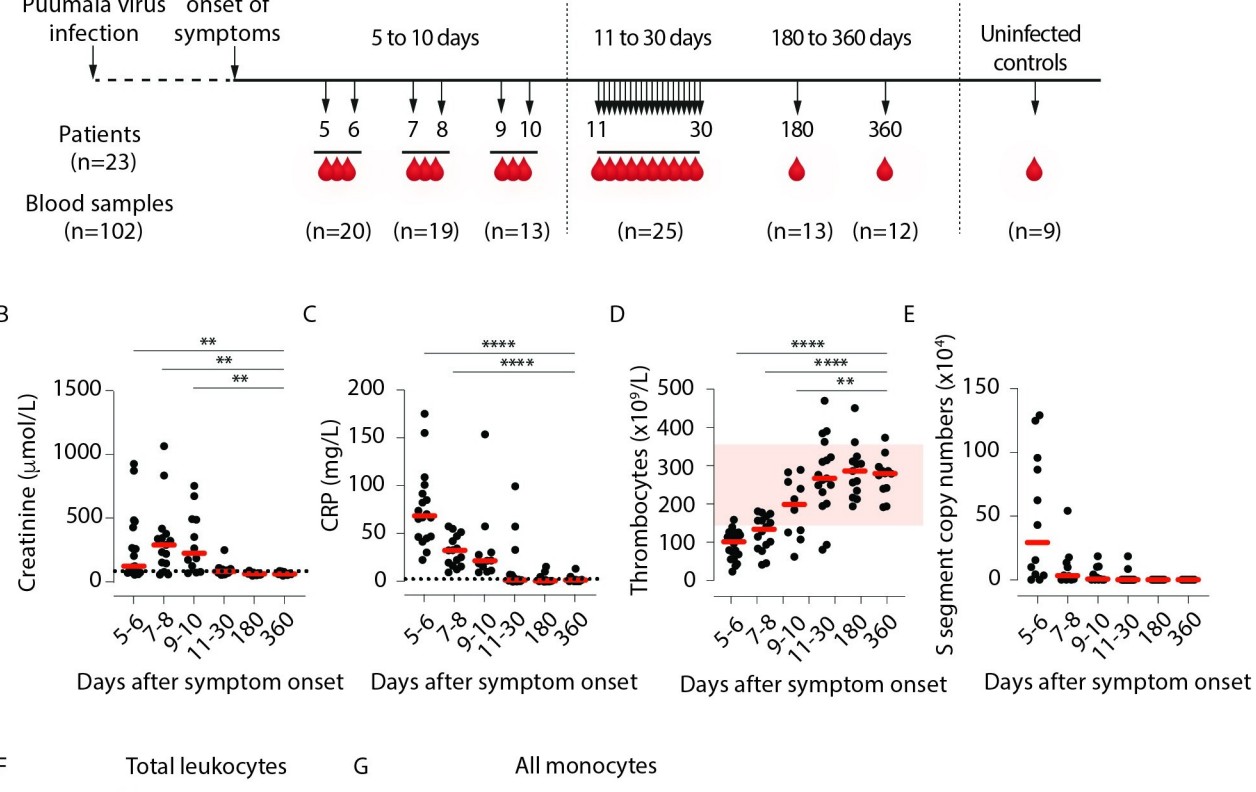

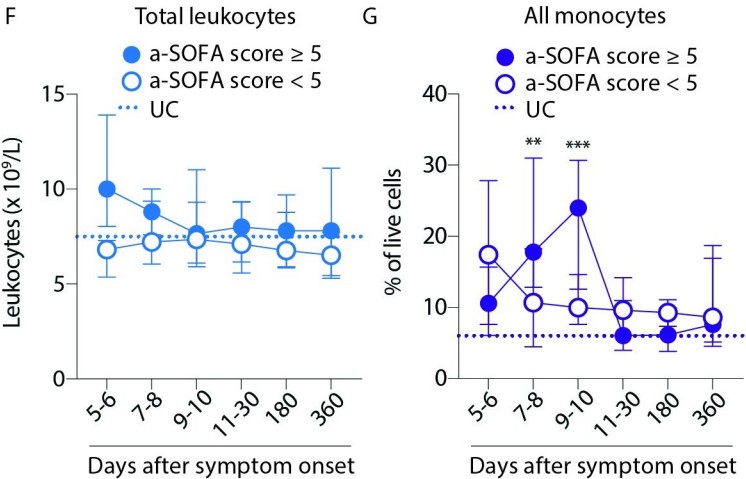

**Fig 1. Viremic phase in Puumala virus-infected patients is characterized by thrombocytopenia, acute kidney injury and an overall increase in monocyte frequencies.** (**A**) Longitudinal peripheral blood samples (n = 102) were collected from 23 patients with serologically-confirmed acute PUUV infection. Samples were obtained from the patients during acute infection (days 5–10) and convalescence (days 11–30, day 180 and day 360) as well as from 9 uninfected controls (UCs). Levels of (**B**) creatinine (μmol/L in plasma), (**C**) C-reactive protein (CRP, mg/L in plasma), (**D**) thrombocyte counts ($10^9$/L in blood) and (**E**) copy numbers of PUUV S segment in plasma ($10^4$/ml in plasma). Graphs show data from individual subjects and lines indicate median values. Dotted lines indicate reference values in healthy adults. Each dot represents an individual sample and red zone indicates reference range for thrombocyte counts in healthy adults (**D**). (**F**) Graph shows the total leukocyte counts (median ± interquartile range, IQR) during PUUV infection for patients stratified by severity as lower, i.e. a-SOFA score <5 (empty circles) or higher, i.e. a-SOFA score ≥5 (filled circles). (**G**) Graph shows median ± IQR monocyte frequencies to total live cells (classical + intermediate + nonclassical monocytes) during PUUV infection for patients stratified by severity as lower, i.e. a-SOFA score <5 (empty circles) or higher, i.e. a-SOFA score ≥5 (filled circles). Dotted line represents median values from UCs. Statistical differences between day 360 and other time points (**B-E**), and patients with lower and higher a-SOFA scores (**F-G**) were assessed using a generalized estimated equation (GEE) model in SPSS and differences were considered significant at p<0.05 (**p<0.01, ***p<0.001 and ****p<0.0001).

**Table 1. Distribution of HFRS patient samples over time.**

| | Days after symptom onset | | | | | | | | | | | | | | | | | | | | | |
|---|---|---|---|---|---|---|---|---|---|---|---|---|---|---|---|---|---|---|---|---|---|---|
| | 5 | 6 | 7 | 8 | 9 | 10 | 11 | 12 | 16 | 17 | 18 | 19 | 20 | 21 | 22 | 23 | 24 | 25 | 26 | 27 | 180 | 360 |
| P01 | ■ | | ■ | | | | | | | | ■ | | | | | | | | | | | |
| P02 | | ■ | | | | | | | | | | | | | ■ | | | | | | ■ | |
| P03 | | | ■ | ■ | | | | | | | | | | | ■ | | | | | | ■ | ■ |
| P04 | | ■ | | | | | | | | | | | | | | | | | | ■ | ■ | |
| P05 | | ■ | ■ | | | | | | | | | | | | | | | ■ | | | | ■ |
| P06 | | ■ | ■ | ■ | ■ | | | | | | | | | | | | | | ■ | | | |
| P07 | | | | ■ | ■ | | | | | | | | | | | ■ | | | | | ■ | ■ |
| P08 | ■ | ■ | ■ | ■ | | | | | | | | | | | | | | | ■ | | | |
| P09 | | ■ | | ■ | | | | | | | | | ■ | | | | | | | | | |
| P10 | | | | | ■ | ■ | ■ | | | | | | | | | | | | ■ | | ■ | ■ |
| P11 | | ■ | | | | | | | | | | | | ■ | | | | | | | ■ | ■ |
| P12 | | ■ | | | | | | | | | | | | ■ | | | | | | | ■ | ■ |
| P13 | | ■ | | ■ | ■ | | | | | ■ | | | | | | | | | | | | |
| P14 | | | | ■ | ■ | | | | | | | | | | | | | | ■ | | ■ | ■ |
| P15 | | | | ■ | ■ | ■ | | | | | | | | | | ■ | | | | | | |
| P16 | | | | ■ | ■ | | | | | | | | | | ■ | | | | | | | |
| P17 | ■ | ■ | | ■ | | | | | | | | | | | ■ | | | | | | ■ | ■ |
| P18 | | | | | | | ■ | ■ | ■ | | | | | | | | | | | | ■ | |
| P19 | ■ | | | ■ | ■ | ■ | | | | | | ■ | | | | | | | | | | |
| P20 | ■ | ■ | | | | | | | | ■ | | | | | | | | | | | | |
| P21 | ■ | ■ | | | | | | | | | | | | ■ | | | | | | | ■ | ■ |
| P22 | ■ | ■ | | | | | | ■ | | | | | | | | | | | | | | |
| P23 | | ■ | | | ■ | | | | | | | | | | ■ | | | | | | ■ | ■ |
| Total (n) | 20 | | 19 | | 13 | | | | | | | | 25 | | | | | | | | 13 | 12 |
| | Acute phase (n = 52) | | | | | | Convalescent phase (n = 50) | | | | | | | | | | | | | | | |

with a-SOFA scores <5 (Fig 1G), suggesting an expansion of leukocytes including monocytes in patients with more severe disease. Interestingly, this coincided with elevated CD34+ progenitor frequencies in blood during acute severe HFRS compared to during convalescence (S1A, S1B and S1C Fig). Taken together, these data suggest that both monocytes and their progenitors are altered during HFRS compared to uninfected controls in a manner that correlates with disease severity.

## Acute PUUV-associated HFRS is characterized by a loss of nonclassical monocytes from circulation

To further dissect the altered total blood monocyte frequencies, we analysed the prevalence of CD14+CD16– classical monocytes, CMs (green); CD14+CD16+ intermediate monocytes, IMs (red) and CD14–CD16+ nonclassical monocytes, NCMs (blue) (Figs 2A and S2A). During the acute phase of HFRS, frequencies of CM and IMs increased compared to uninfected controls (UCs), but returned to similar frequencies as in UCs during convalescence (Fig 2B and 2C). In contrast, NCMs were significantly reduced in frequency during acute HFRS compared to convalescence (Fig 2D). In the DC compartment, no statistically significant fluctuations in frequencies over course of disease was observed (S3 Fig). When we compared the monocyte frequencies of patients stratified by a-SOFA scores, we observed that in the patients with higher a-SOFA scores, peak frequencies of CMs and IMs were observed later as compared to

**Table 2. Clinical and laboratory characteristics of HFRS patients.**

| Patient ID | Age (years) | Sex | Smoker | Days in hospital | CRP* (mg/L) | Leukocytes~ ($10^9$/L) | Thrombocytes§ ($10^9$/L) | Creatinine¶ (µmol/L) | Mean arterial pressure# (mmHg) | a-SOFA score† |
|---|---|---|---|---|---|---|---|---|---|---|
| P01 | 34 | F | Yes | 4 | 70.5 | 9.8 | 96 | 67 | 70 | 2 |
| P02 | 38 | F | Yes | 8 | 109.1 | 9.6 | 55 | 91 | 79 | 5 |
| P03 | 41 | F | Yes | 4 | 15.9 | 8.9 | 156 | 68 | 87 | 2 |
| P04 | 41 | F | Yes | 4 | 30.5 | 9.5 | 106 | 63 | 85 | 2 |
| P05 | 46 | F | No | 6 | 66.9 | 11.5 | 23 | 240 | 68 | 6 |
| P06 | 56 | F | No | 8 | 25.5 | 6.9 | 118 | 297 | 98 | 5 |
| P07 | 56 | F | Yes | 4 | 30.5 | 9.2 | 83 | 78 | 89 | 3 |
| P08 | 57 | F | Yes | 11 | 101.1 | 9.7 | 124 | 491 | 66 | 8 |
| P09 | 25 | M | No | 4 | 22.8 | 9.8 | 139 | 878 | 85 | 5 |
| P10 | 33 | M | Previous | 7 | 15.9 | 7.4 | 212 | 679 | 91 | 7 |
| P11 | 37 | M | Yes | 5 | 55.3 | 14.8 | 38 | 384 | 84 | 6 |
| P12 | 40 | M | Yes | 8 | 32.8 | 12.2 | 43 | 1071 | 92 | 7 |
| P13 | 42 | M | Yes | 7 | 45.3 | 13.6 | 75 | 480 | - | 6 |
| P14 | 43 | M | Yes | 9 | 57.8 | 12.9 | 257 | 359 | 90 | 6 |
| P15 | 48 | M | Previous | 8 | 47.5 | 14.5 | 101 | 841 | 103 | 5 |
| P16 | 51 | M | Yes | 4 | 42.4 | 7.5 | 93 | 85 | 85 | 3 |
| P17 | 52 | M | Yes | 3 | 91.9 | 6.8 | 110 | 152 | 92 | 2 |
| P18 | 52 | M | Previous | 9 | 153.7 | 12.1 | 62 | 117 | 68 | 4 |
| P19 | 54 | M | No | 8 | 85.2 | 16.9 | 56 | 349 | 77 | 5 |
| P20 | 55 | M | Previous | 3 | 67 | 5.4 | 125 | 81 | 90 | 1 |
| P21 | 63 | M | No | 3 | 175.3 | 8.7 | 79 | 125 | 86 | 3 |
| P22 | 64 | M | No | 3 | 82.5 | 5.2 | 68 | 82 | 79 | 2 |
| P23 | 65 | M | Yes | 6 | 19.2 | 7.2 | 78 | 263 | 85 | 4 |

F = female, M = male. Acute phase laboratory results represent highest values for CRP, creatinine, leukocytes and lowest values for thrombocytes and mean arterial pressure.

\* Plasma C-reactive protein (CRP); reference <3 mg/L.

~ Leukocyte count; normal range 3.4–8.2 x$10^9$/L.

§ Thrombocyte count; normal range 150–360 x$10^9$/L.

¶ Plasma creatinine; reference <90 µmol/L for women, <100 µmol/L for men.

# Mean arterial pressure; reference 70–100 mmHg.

† Adapted-sequential organ failure assessment (a-SOFA) score; calculated from thrombocytes, creatinine and mean arterial pressure.

patients with lower a-SOFA scores (peak frequencies observed at 5–6 days after symptom onset) (Fig 2E and 2F). Strikingly, we observed a pronounced drop in NCM frequencies in both groups early in the acute phase, that remained low also during convalescence compared to UCs (Fig 2G).

## Monocytes are found in increased numbers in the kidneys of patients with acute PUUV-HFRS

We stained a unique set of bouin-fixed paraffin-embedded kidney biopsies from 37 patients with acute HFRS and 43 patients with other kidney diseases (controls) for HLA-DR, CD14, CD16 and CD68 (Fig 3A–3D). The frequency of HLA-DR+, CD14+, CD16+ and CD68+ cells were all significantly increased in patients with HFRS as compared to patients with other kidney diseases (Fig 3E–3H). The cells, likely monocyte-derived macrophages, localized predominantly to tubulointerstitial space (Fig 3A–3D), consistent with the typical finding of

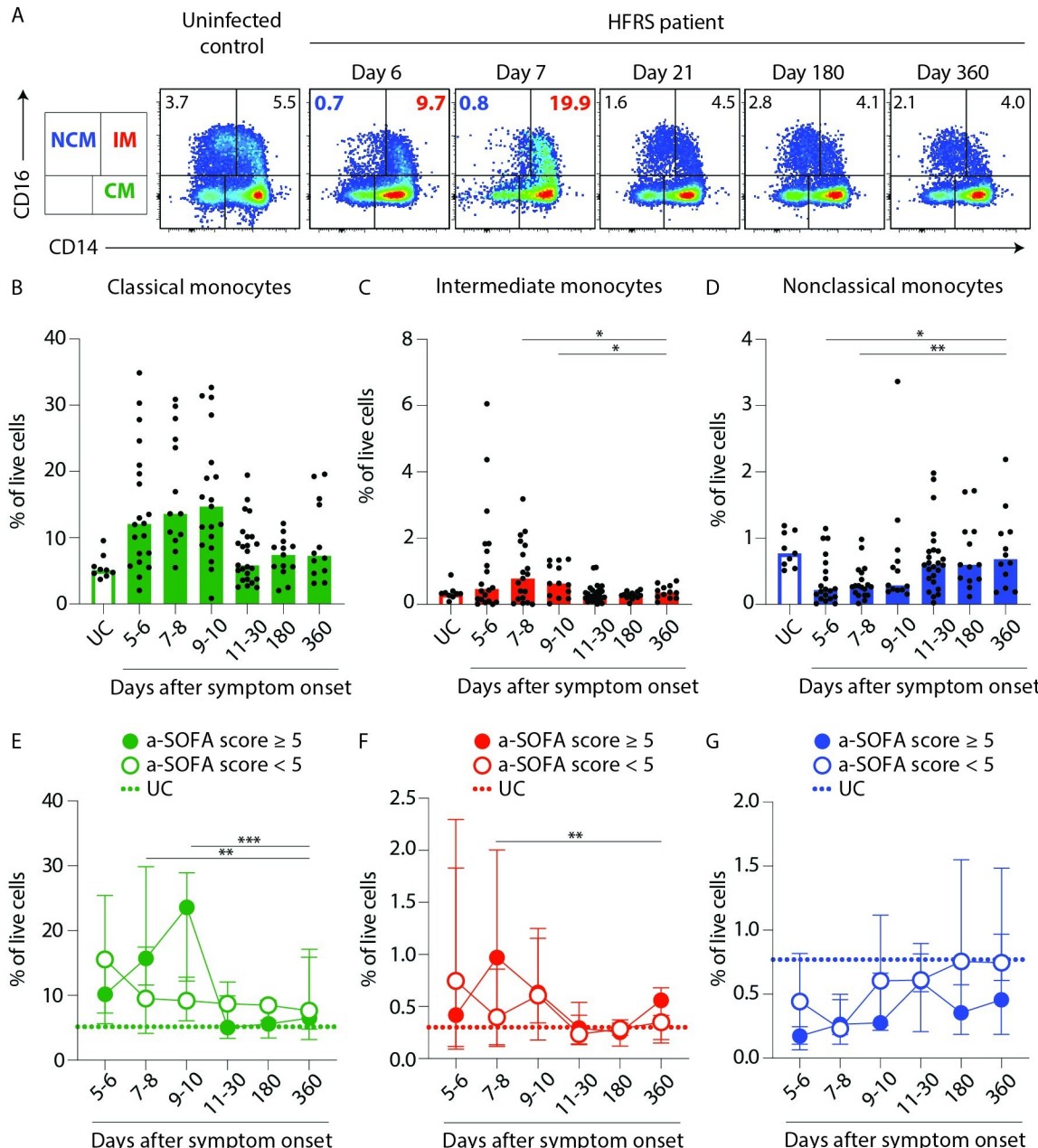

**Fig 2. Sustained depletion of nonclassical monocytes in blood from patients with severe HFRS as compared to uninfected controls.** (**A**) Plots depict the monocyte subsets in a UC and a representative HFRS patient over the course of disease (day 6–360) (classical (CD14 +CD16−), intermediate (CD14+CD16+) and nonclassical (CD14−CD16+) monocytes). (**B-D**) Graphs show frequencies (to total live cells) of (**B**) CMs (green), (**C**) IMs (red) and (**D**) NCMs (blue) in PBMCs from patients (filled bars) and HCs (empty bars). Circles represent individual patients. Statistical differences between day 360 and other time points were assessed using a generalized estimated equation (GEE) model in SPSS and differences were considered significant at p<0.05 (*p<0.05 and **p<0.01). (**E-G**) Graphs show median ± IQR frequencies of (**E**) CMs (green), (**F**) IMs (red) and (**G**) NCMs (blue) in PBMCs from patients stratified by severity as lower, i.e. a-SOFA score <5 (empty circles) or higher, i.e. a-SOFA score ≥5 (filled circles). Median frequencies of respective monocyte subset in UCs are indicated with dotted lines. Statistical differences between patients with lower and higher a-SOFA scores were assessed using a generalized estimated equation (GEE) model in SPSS and differences were considered significant at p<0.05 (**p<0.01 and ***p<0.001).

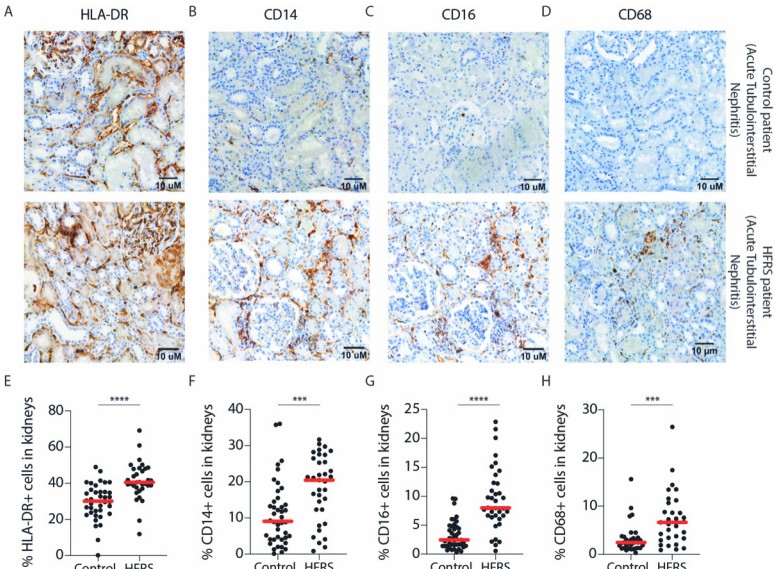

**Fig 3. Increased numbers of CD14+ and CD16+ cells are detected in kidneys during HFRS. (A-C)** Representative images of HRP-based kidney immunohistochemistry for (**A**) HLA-DR, (**B**) CD14 and (**C**) CD16 and (**D**) CD68 in control (top panels) and acute HFRS patients (lower panels), both patients diagnosed with acute tubulointerstitial nephritis. Hematoxylin was used as the counterstain. (**E-H**) Graphs show percentages of (**E**) HLA-DR (**F**) CD14 (**G**) CD16 and (**H**) CD68 positive cells out of all cells in 43 non-HFRS controls and 37 acute HFRS kidney tissues as assessed by automated computing methods. Statistical differences between groups were assessed by Mann-Whitney test and considered significant at p<0.05. (***p <0.001 and ****p< 0.0001).

tubulointerstitial nephritis associated with acute HFRS (S2 Table). In addition, HLA-DR+ positive cells were found also in the glomeruli, which due to the absence of CD14, CD16 or CD68 most likely represented cells not from the monocyte lineage. The infiltration of monocytes and monocyte-derived cells into the kidneys during HFRS may contribute to the renal pathology observed during HFRS. We also noted that while the majority of monocytes detected in the kidneys during HFRS were predominantly CD14+, the high frequencies of CD16+ could explain the loss of NCMs from circulation. In addition, sustained mobilization of CD14+ CMs to renal tissues during HFRS (as seen in the biopsies) may contribute to decreased differentiation of CMs to IMs to NCMs in circulation.

## Maturation and activation of monocytes during acute PUUV-HFRS

To understand whether alterations in monocyte frequencies were related to activation of the cells during HFRS, we determined the cell surface expression of maturation, migration and endothelial adhesion markers on the blood monocyte subsets (S2B Fig). All three monocyte subsets displayed significantly increased HLA-DR during acute HFRS compared to UCs that normalized during late convalescence (Fig 4A). The CD16-expressing IMs and NCMs significantly upregulated CCR2 (Fig 4B) and CD62L (Fig 4C) expression during acute PUUV-HFRS, as compared to late convalescence. And finally, CCR7 was upregulated on all three monocyte subsets during acute disease (Fig 4D). Interestingly, we found significantly stronger upregulation of CCR7 on CMs and IMs in patients with lower a-SOFA scores as opposed to those with higher a-SOFA scores (Fig 4E–4G). We also noted that CCR2 was upregulated to a greater extent by blood IMs in patients with higher a-SOFA scores (although statistically non-significant; Fig 4H). As kidneys are affected during HFRS, we hypothesized that in response to PUUV infection, monocytes may upregulate migration markers, infiltrate the kidneys and contribute to the renal symptoms documented in HFRS patients.

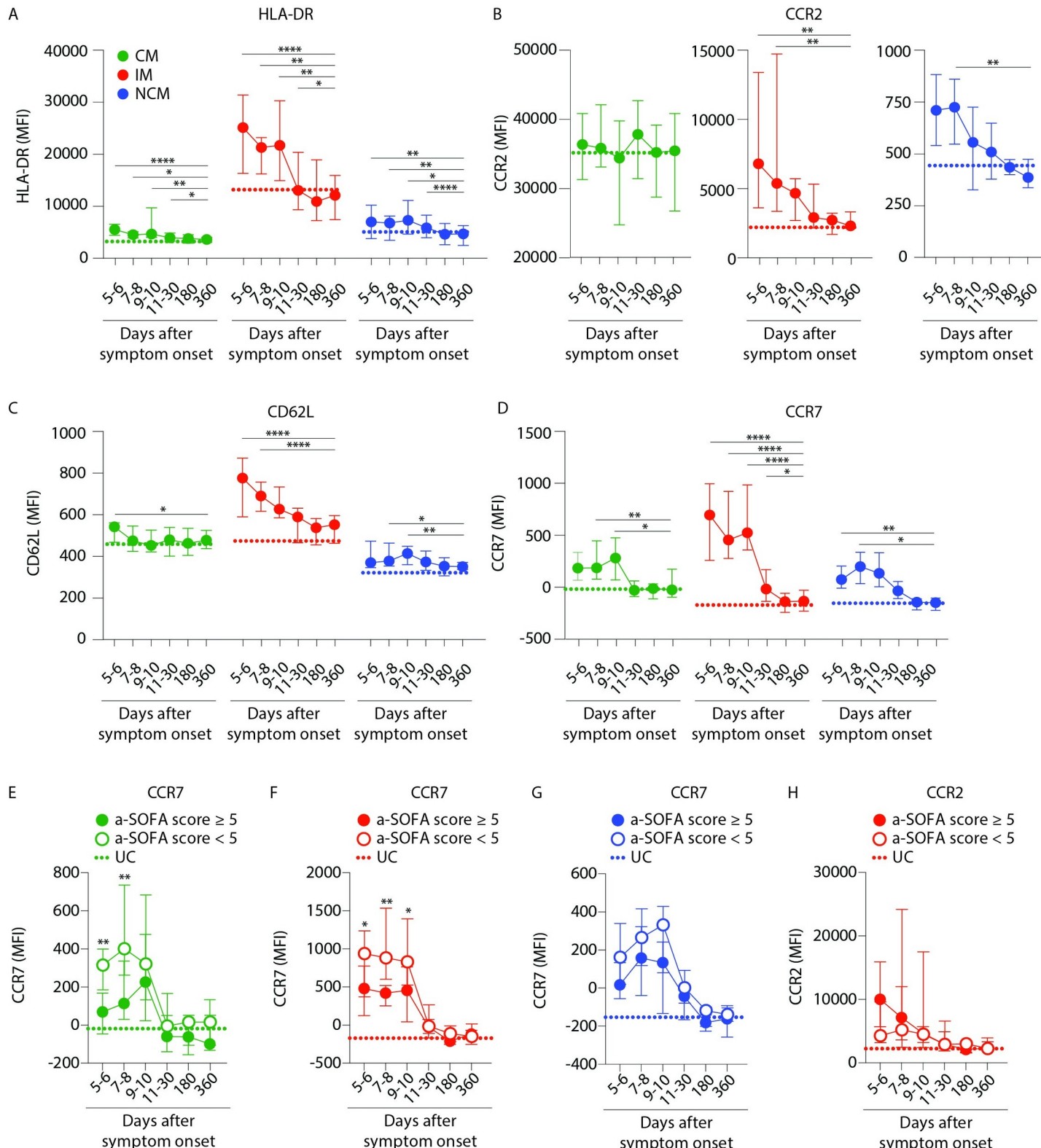

**Fig 4. Intermediate and nonclassical monocytes are more activated and express more endothelial adhesion and migration markers during acute HFRS. (A-D)** Graphs show median ± IQR MFI of (**A**) HLA-DR, (**B**) CCR2 (**C**) CD62L and (**D**) CCR7 surface expression on monocyte subsets (CM- green, IM-red and NCM- blue) in PBMCs over the course of HFRS. Median MFI values from UCs are displayed as dotted lines for each monocyte subset. Statistical differences between day 360 and other time points were assessed using a generalized estimated equation (GEE) model in SPSS and differences were considered significant at p<0.05 (*p<0.05, **p<0.01,

[***]p<0.001 and [****]p<0.0001). (**E-H**) Graphs show median ± IQR MFI of CCR7 expression on (**E**) CMs, (**F**) IMs and (**G**) NCMs; and (**H**) MFI of CCR2 expression on IMs from patients stratified by severity as lower, i.e. a-SOFA score <5 (empty circles) or higher, i.e. a-SOFA score ≥5 (filled circles). Median frequencies of respective monocyte subset in UCs are indicated with dotted lines. Statistical differences between patients with lower and higher a-SOFA scores were assessed using a generalized estimated equation (GEE) model in SPSS and differences were considered significant at p<0.05 ([*]p<0.05 and [**]p<0.01).

## CD11c+ myeloid cells display impaired TNFα-production during acute PUUV-HFRS

To determine whether HFRS also affected direct functional properties of blood monocytes, we assayed the ability of CD11c+ myeloid cells to produce the key proinflammatory cytokines IL-6 and TNFα with or without TLR7/8 stimulation. This required in vitro culture and stimulation of the cells, which we optimized to 3 hours to minimize effects on cell surface proteins used to identify cells of interest. Still, even 3 hours of culture resulted in downregulation of CD16 [29], which prevented gating on separate monocyte subsets. Instead, we analysed the cytokine response in total CD11c+ myeloid cells in the patient samples (S4A Fig). In the earliest acute phase of HFRS, a small proportion of the CD11c+ cells produced IL-6 without TLR stimulation, which gradually disappeared over the course of disease (S4B and S4C Fig). Upon TLR7/8L stimulation, 2–5% of the CD11c+ myeloid cells expressed IL-6 in both UC and acute and convalescent patient samples but we did not observe significant differences in their response to TLR stimulation (S4D Fig). IL-6 production in the absence of TLR stimulation negatively correlated with thrombocytopenia, but only in patients with higher a-SOFA scores (S4E Fig). In contrast to IL-6, the frequency of TNFα producing CD11c+ cells did not vary significantly across the course of HFRS, or between patients and UCs (Fig 5A). However, we observed a distinct impairment in the TNFα production from CD11c+ myeloid cells in response to TLR7/8L stimulation, during the acute phase as compared to UCs as well as at day 360. Additionally, in patients with higher a-SOFA scores, the impairment in TNFα production correlated with thrombocytopenia over the course of disease (Fig 5B). Taken together, these

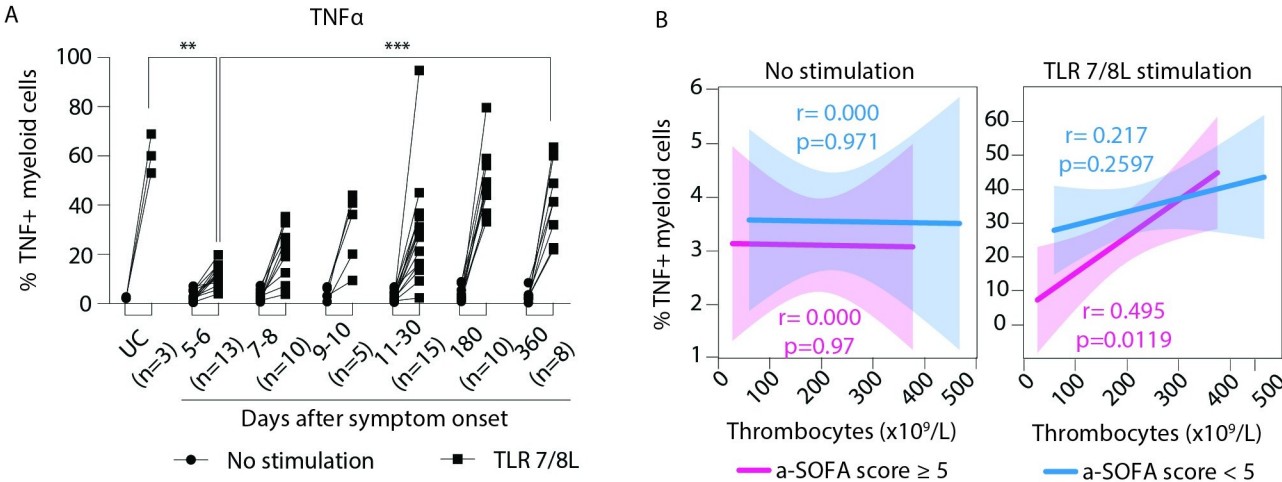

**Fig 5. Myeloid cells from acute HFRS patients produce IL-6 but have impaired TNFα production in response to additional TLR stimulation. (A)** Graph displays median frequency ± IQR TNFα producing cells in CD11c+ myeloid cells in PBMCs in UCs (n = 3) and HFRS patients (n = 15) in the absence (circle) or presence (square) of TLR 7/8L stimulation for 3h. Statistical differences between groups (of similar exposure conditions) were assessed by mixed-effects analysis using Dunnett's multiple correction test and considered significant at p< 0.05. (**p<0.01 and ***p<0.001). (**B**) Lines display bivariate linear regression analysis between thrombocyte counts ($10^9$/L) and frequency of (**C**) IL-6 or (**D**) TNFα producing CD11c+ myeloid cells without (left) or with (right) TLR 7/8L stimulation. The shaded area represents the 95% confidence region for the fitted line. Patients are stratified by severity as lower, i.e. a-SOFA score <5 (blue) or higher, i.e. a-SOFA score ≥5 (pink). r represents Spearman ρ and statistical differences were considered significant at p<0.05. UC = uninfected controls.

data suggest not only an alteration in frequency but also in certain functional responses within the myeloid immune cell compartment during HFRS.

## Cytokine levels were elevated in plasma during acute PUUV-HFRS

The functional properties of monocytes are strongly influenced by cytokines and chemokines present in the local environment. To obtain insights on the inflammatory and pro-migratory milieu in peripheral blood during acute HFRS, we measured the levels of various cytokines and chemokines in plasma samples that matched the analysed PBMC samples for patients and time points. We measured the concentrations of the cytokines TNFα, IL-6, IL-18 IL-10, IL-1β, IL-12p70, IFNα, IFNγ, M-CSF and GM-CSF; and the chemokines CCL2, CCL3 and CCL7. In general, several cytokine levels (TNFα, IL-6, CCL2, IL-8, IL-10 and IL-18) were mildly elevated during the acute phase (days 5–10) as compared to UCs, and declined to steady state levels in convalescence (Fig 6A). Somewhat surprisingly, when split by a-SOFA severity score, there was no consistent pattern in the plasma concentrations from patients with higher vs. lower scores (Fig 6B–6E). The proinflammatory cytokines IL-6 (Fig 6B) and TNFα (Fig 6C), but also the more immunoregulatory cytokine IL-10 (Fig 6D) were elevated in patients compared to UCs. Mild cases displayed a second peak of these cytokines around day 9–10 which was not seen in patients with severe disease. The chemokine CCL2 was only moderately upregulated compared to UCs (Fig 6E). We observed a correlation between cytokine levels and thrombocytopenia (Fig 6F); and between cytokine levels and plasma creatinine levels (Fig 6G), both hallmarks of acute PUUV-associated HFRS suggesting that, although not supported by a-SOFA based patient stratification, cytokine production in patients may still contribute to HFRS disease. We also detected a significant increase in the levels of M-CSF in patients during acute HFRS as compared to convalescent phase (Fig 6H) supporting the idea of active myelopoiesis during acute HFRS.

## CD16+ monocytes show more pronounced CD62L upregulation in response to direct virus exposure in vitro

During acute PUUV-HFRS, monocytes in circulation are directly exposed to the virus as well as to virus-infected endothelial cells. Since we observed increased expression of the endothelial adhesion marker CD62L on CD16-expressing monocytes (IMs and NCMs) and CCR7 on all monocyte subsets in patients with acute HFRS caused by PUUV, we investigated the consequences of in vitro exposure to PUUV-Suonenjoki strain (shown to have wild-type properties [30]) of bead-enriched CD16+ (IMs and NCMs) and CD16– (CMs) monocytes (Fig 7A). Monocytes were either directly exposed to virus or added to a monolayer of PUUV-infected blood microvascular endothelial cells (BECs). Both CD16+ and CD16– monocytes upregulated CD62L after direct exposure to PUUV or TLR7/8L, but in both cases CD16+ monocytes displayed higher level of CD62L induction than CD16– monocytes (Fig 7B, top panel). This finding suggests that CD16+ monocytes respond more strongly to hantaviruses than CD16– monocytes, which is in line with previous studies showing the ability of NCMs to efficiently sense viruses. Interestingly, after exposure to BECs, regardless of infection or TNF-α treatment, both CD16+ and CD16– monocytes lost their CD62L expression, possible due to binding with CD34, the endothelial cell ligand for CD62L, and subsequent cleaving and/or internalization.

CD16– monocytes consistently displayed higher CCR7 expression than CD16+ monocytes after direct exposure to PUUV as well as after co-incubation with PUUV-infected BECs (Fig 7B middle panel). Strikingly, while direct exposure of CD16– or CD16+ monocytes to virus or TLR7/8L did not have a clear impact on CCR7 expression, exposure to infected or TNFα

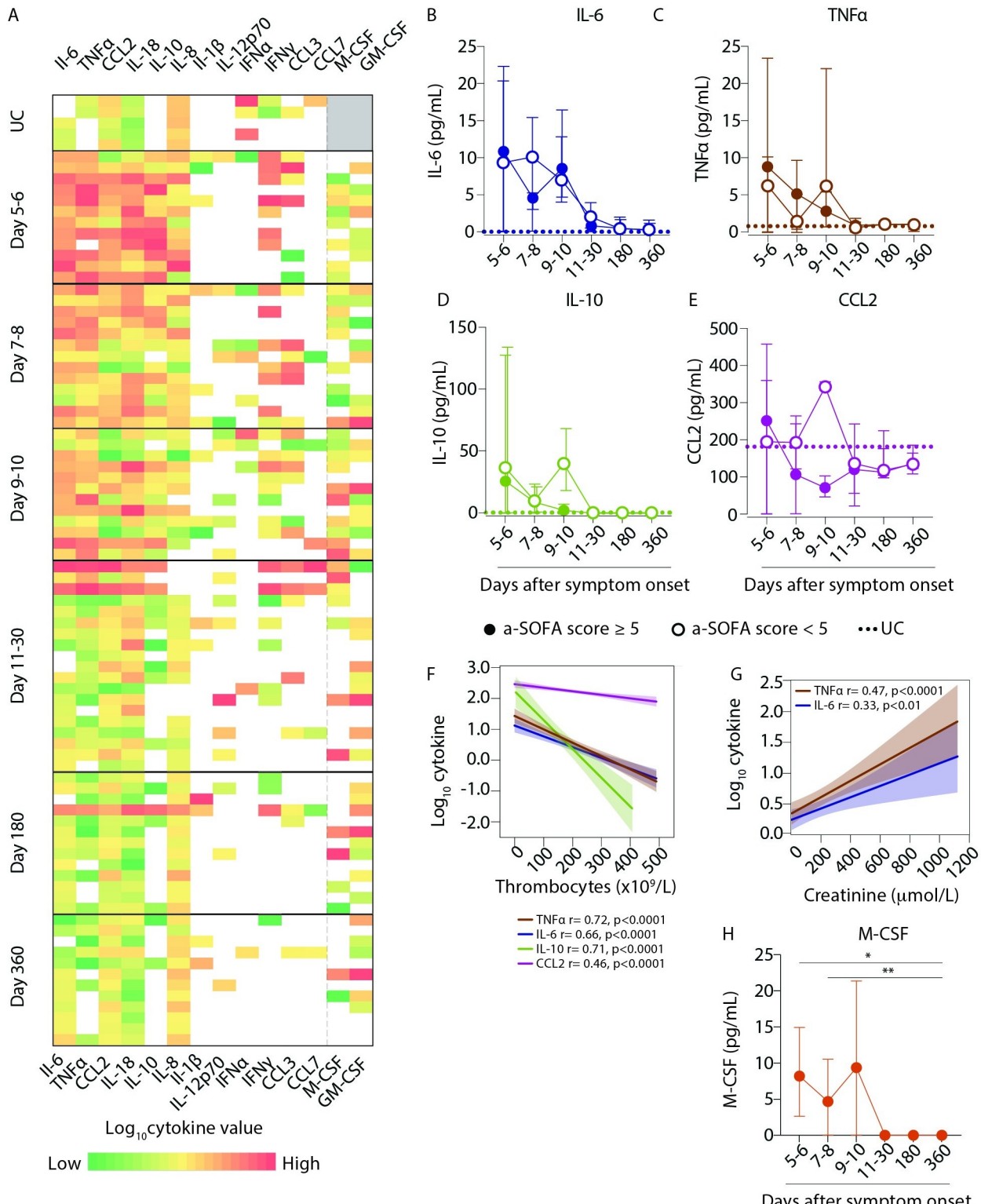

**Fig 6. Plasma levels of pro-inflammatory cytokines are elevated during HFRS infection.** (**A**) Heatmap represents log₁₀ cytokine concentrations (as assayed by Luminex and ELISAs) in plasma for UCs (n = 5) and HFRS patients (n = 21) over the course of acute PUUV infection and during convalescence. The color bar ranges from min (green) to max (red) within a column for each cytokine, yellow representing median values and white boxes representing no cytokine detection, and grey represents no samples assayed. (**B-D**) Graphs show median ± IQR plasma concentrations (pg/mL) of (**B**) IL-6, (**C**) TNFα, (**D**) IL-10 and (**E**) CCL2 in patients stratified by severity as lower, i.e. a-SOFA score <5 (empty circles) or higher,

i.e. a-SOFA score ≥5 (filled circles). Median plasma cytokine concentration in UCs are indicated with dotted lines. (**F-G**) Lines display bivariate linear regression analysis between $\log_{10}$ cytokine values of (**F**) TNFα, IL-6, IL-10 and CCL-2 against thrombocyte counts ($10^9$/L). (**G**) Lines of fit display bivariate linear regression analysis between $\log_{10}$ cytokine values of TNFα and IL-6 against creatinine levels (μmol/L). The shaded area represents the 95% confidence region for the fitted line. r represents squared Spearman ρ and statistical differences were considered significant at $p < 0.05$. UC = uninfected controls. (**H**) Graph shows median ± IQR plasma concentrations (pg/mL) of M-CSF over the course of HFRS in patients. Statistical differences between day 360 and other time points were assessed using a generalized estimated equation (GEE) model in SPSS and differences were considered significant at $p < 0.05$ (*$p < 0.05$ and **$p < 0.01$).

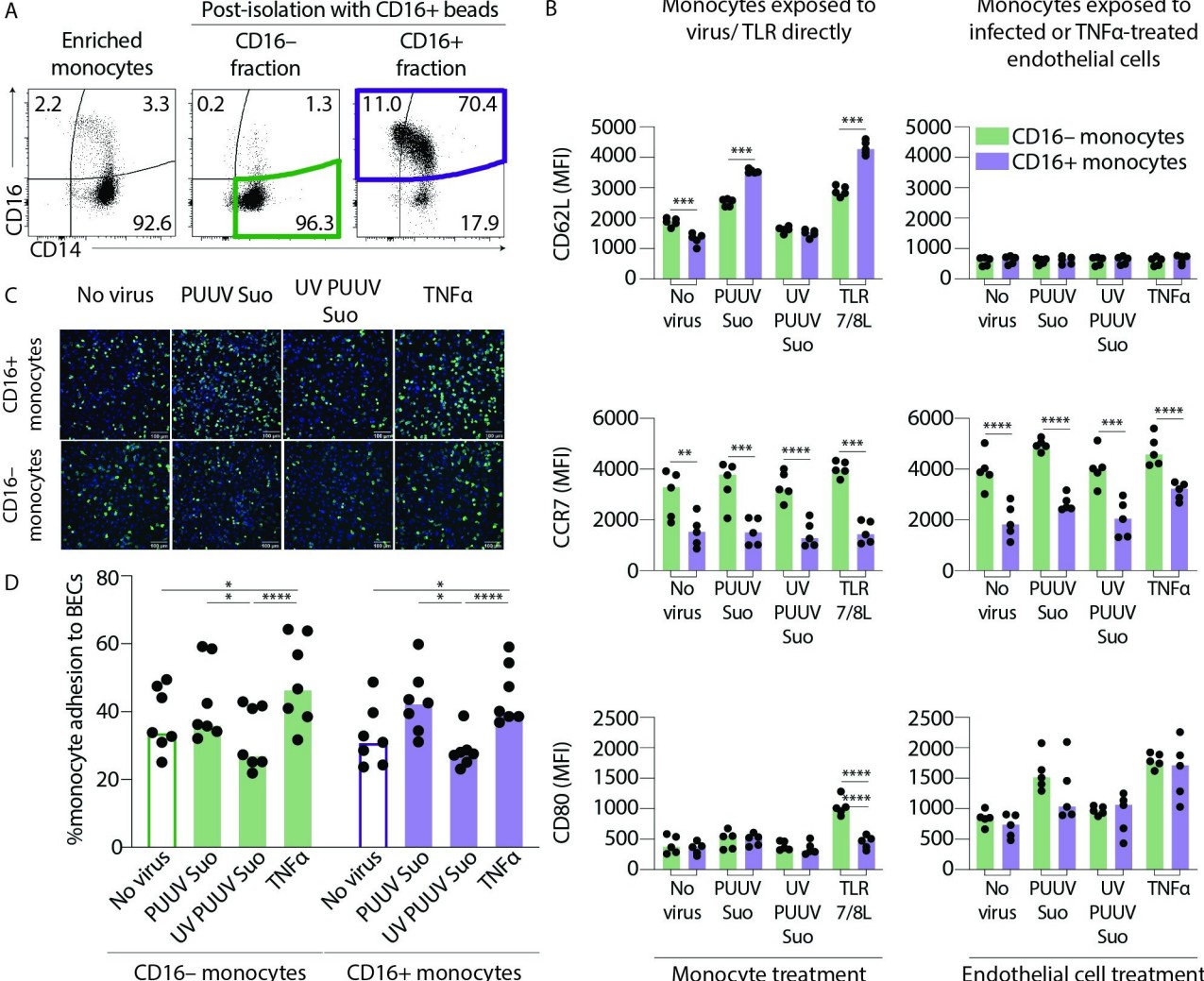

**Fig 7. CD16+ and CD16– enriched monocytes respond differently during in vitro PUUV infection.** (**A**) Peripheral blood monocytes were first enriched by negative selection to remove B, T, NK cells and granulocytes using a RosetteSep enrichment kit. Plots show a representative sample of enriched monocytes before (pre-isolation) and after (post-isolation) isolation using magnetic CD16+ beads. Post-isolation graphs show CD16– fraction that is enriched in CD14+ CMs (green gate) and the CD16+ fraction that is enriched in CD16+ IMs and NCMs (violet gate). (**B**) Graphs show MFI of CD62L (top panels), CCR7 (middle panels) and CD80 (lower panels) expression on CD16– (green) or CD16+ (violet) monocytes exposed to PUUV (MOI = 1) directly for 1 h followed by 19 hr of incubation with fresh media (left panels) or PUUV-infected (MOI = 10, for 20 h) BECs (right panels); n = 5 from two independent experiments. Statistical differences between CD16+ and CD16– monocytes were assessed by two-way ANOVA with Tukey's multiple comparisons test and considered significant at $p < 0.05$ (**$p < 0.01$, *** $p < 0.001$ and ***$p < 0.0001$). (**C**) Representative images of HLA-DR staining of CD16+ (top panels) and CD16– (lower panels) monocytes adhered to endothelial cells (n = 7 donors). HLA-DR staining (green) defines monocytes and endothelial cell nuclei are counterstained with Hoechst 3342 (blue). (**D**) Graph summarizes % adherence of CD16+ (violet) or CD16– (green) monocyte adhesion to endothelial cells and lines connect data from same donors. Statistical differences were assessed by two-way ANOVA with Tukey's multiple comparisons test and considered significant at $p < 0.05$.

treated BECs upregulated CCR7 expression in both cell types, suggesting that activated endothelial cells but not the virus itself are needed for monocyte CCR7 upregulation in vitro, and perhaps in HFRS patients. We did not observe any significant differences in CD80 upregulation between CD16– and CD16+ monocytes, except upon direct exposure to TLR7/8L (Fig 7B, bottom panel). We also assessed the ability of CD16+ and CD16– monocytes to adhere to PUUV-infected BECs by confocal imaging and quantification (Fig 7C). Both CD16+ and CD16– monocytes displayed increased adhesion to PUUV-infected or TNFα-treated BECs suggesting that PUUV infection elevates the adhesive properties of endothelial cells towards monocytes (Fig 7D). This mechanism might explain the initial loss of monocyte subsets from circulation during acute HFRS.

## Discussion

HFRS and HPS are characterized by vascular leakage due to increased endothelial permeability. Circulating monocytes, especially patrolling NCMs, are in constant interaction with the endothelium and may play important roles in hantavirus pathogenesis. We report that the frequencies of circulating NCMs significantly decreased whereas CMs and IMs increased during the acute stage of HFRS. In line with this, we also observed increased monocyte infiltration into kidney tissues of patients with acute HFRS. In addition to increased HLA-DR expression on all monocyte subsets during acute HFRS, we observed significant upregulation of CD62L during acute HFRS, suggesting increased adhesion to the endothelium and migration into peripheral tissues. Patients with higher a-SOFA scores (indicative of severe disease), displayed significantly different kinetics for monocyte frequencies in circulation, as well as migration markers, than patients with lower scores. Increased endothelial adhesion and infiltration to the kidneys may together explain the observed differences in blood monocyte frequencies and may contribute to the renal pathology during acute HFRS.

In line with the current study, increased numbers of circulating CMs and IMs as well as decreased NCMs have been observed for Hantaan virus (HTNV) caused HFRS [31]. However, in contrast to the current study, the loss of NCMs during HTNV-caused HFRS associated with favourable disease outcome whereas increased CMs and IMs positively correlated with higher disease severity. Therefore, the frequency of the different monocyte subsets in circulation during HFRS may also depend on the virus causing HFRS. Interestingly, the loss of NCMs associated with fatality in acute Ebola virus infection [32] implying that loss of monocytes from circulation can strongly influence the pathogenesis and outcome of viral hemorrhagic fevers in general. In our prior study of PUUV-caused HFRS, performed on patients sampled in Northern Sweden, we observed diminished absolute numbers of all monocyte and DCs subsets during the acute stage of the disease [33]. The underlying cause of the different patterns of monocyte and DC responses observed between our two studies on PUUV-caused HFRS patients may stem from geographical differences. Altogether, the clinical severity of HFRS caused by PUUV; the severity of AKI (plasma creatinine elevation), inflammation, thrombocytopenia and capillary leakage, may suggest that HFRS is more severe in the Finnish cohort as compared to the previously-analysed Swedish cohort [33–35]. This disparity in disease severity may be attributed to, differences in host genetic factors or distinct PUUV strains circulating in Northern Sweden and Central Finland [36]. Certain HLA-B and HLA-DR haplotypes have been associated with more severe disease following infection with old and new world hantaviruses, and polymorphisms in the TNF, VE-cadherin, integrin and endothelial nitric oxide synthase genes have been associated with increased risk of severe disease in HFRS [37–39]. Although highly speculative, the apparent increase in monocyte and DC frequencies towards the Finnish PUUV strain as opposed to the one circulating in Northern Sweden could also be

attributed to the different strains. In any case, further work is needed to clarify potential differences in HFRS disease manifestation between the two geographical locations.

The patients analysed in this study exhibited some upregulation of cytokines (TNFα, IL-6, IL-18 and IL-10) during the acute stage of the disease compared to uninfected controls, which gradually diminished to normal levels during late convalescence. This is in line with the acute inflammation previously described in non-fatal hantavirus infections [40,41]. As expected [42], increased levels of the pro-inflammatory cytokine TNFα associated with both severity of AKI (plasma creatinine elevation) and thrombocytopenia. Additionally, elevated levels of IL-10 at early acute stage associated with milder disease, indicating a possible role for the broad anti-inflammatory and immunoregulatory functions of IL-10 [43]. We observed an increased frequency of IL-6 producing CD11c+ cells, which implies that myeloid cells contribute to increased plasma IL-6 levels during acute HFRS. However, this was not the case for TNFα, for which CD11c+ cells from the acute stage of the disease showed impaired expression levels upon in vitro challenge by TLR7/8 ligand, suggesting dysregulation of the myeloid cell compartment due to continued immune challenge during acute HFRS. The CD11c+ cells, largely CD14+ classical monocytes, stimulated in vitro with TLR7/8L for 3 hours, also displayed downregulation of CD16. The brief duration of this experiment restricted our ability to study the exact phenotype of these cells in greater detail as previously reported [44]. To understand the differentiation capacity of the monocytes remaining in circulation, future in vitro studies including trajectory analysis may provide additional insight. CCL2 levels in plasma were moderately upregulated during acute HFRS, despite higher expression of its cognate receptor CCR2 on IMs and NCMs in HFRS patients compared to UCs. In our previous study [33] we observed downregulation of CCR2 by monocytes exposed to PUUV in vitro, which is in contrast to our observations in HFRS patients. The CMs we observed in circulation may not have been exposed to the virus, whereas monocytes that did encounter the virus may have migrated to the kidneys, as suggested by the kidney biopsy data. The disparity in observations between ex vivo and in vitro studies may be aided by chemokine receptors being quite sensitive to culture conditions, and by different viral strains being investigated in the studies. CCL2 is a well-documented chemoattractant for monocytes and promotes cell migration towards sites of inflammation [45,46]. However, CCL2 also facilitates monocyte egress from the bone marrow [22,47]. As suggested by increased frequencies of CMs, CD34+ hematopoietic progenitors as well as elevated levels of CCL2 and M-CSF, it is likely that emergency myelopoiesis occurs in the bone marrow of patients with acute HFRS to replenish the early loss in monocytes and overall loss of NCMs. This may result in increased frequencies of circulating monocyte precursors later during acute disease and subsequently, normalization of monocyte frequencies to levels seen in UCs during the convalescent phase of HFRS, and merits further investigation in future studies.

All the monocyte subsets exhibited significantly elevated cell surface expression of HLA-DR during the acute stage of HFRS, particularly IMs, which is in line with their superior ability among monocytes to evoke T cell responses [48]. Our findings also show increased levels of CCR7 on all monocyte subsets during acute HFRS, which in the case of CMs and IMs associated with lower disease severity. CCR7 regulates trafficking of leukocytes to and within the lymphatic system [49]. In CCR7-/- mouse models, circulating leukocytes are significantly increased and elevated numbers of monocytes migrate into tissues where they can cause damage [50,51]. This observation together with our previous findings suggest that in acute HFRS patients with higher monocyte CCR7 levels, CCR7 may regulate monocyte migration to lymph nodes instead of unchecked peripheral tissue infiltration and possibly, subsequent immune-mediated damage. In addition, stronger CCR7-mediated homing to lymph nodes will likely prime adaptive immune responses more efficiently, which may again contribute to the milder

pathology observed. Conversely, patients with low CCR7 expression may have increased monocyte migration to peripheral tissues rather than lymph nodes, contributing to the pathology associated with HFRS, as suggested previously [52]. Since we found increased numbers of monocytes and monocyte-derived cells in the kidneys during HFRS as compared to other kidney diseases, monocyte extravasation to the kidneys from circulation during acute HFRS could help explain the loss of NCMs from blood, and the presence of CD68-expressing macrophage-like cells suggests in situ monocyte differentiation into macrophages.

To identify the manner of monocyte subset activation during HFRS, blood CD16+ (IMs and NCMs) and CD16– (CMs) monocytes were isolated and exposed directly to PUUV or to PUUV-infected primary endothelial cells, evaluating CD62L and CCR7 expression in particular. When directly exposed to PUUV, stronger upregulation of CD62L was observed on IMs and NCMs than on CMs, in line with our observations in patients. CD62L expression was lost on all monocytes in contact with endothelial cells, suggesting adhesion-mediated changes in monocyte CD62L expression. Upregulation of CCR7 by monocytes was only observed when exposed to PUUV-infected endothelial cells but not when directly exposed to virus, and consistently higher in CMs. Therefore, contact with infected endothelial cells may be required for CCR7 upregulation on monocytes in acute HFRS, the magnitude positively correlating with milder disease. It is unlikely that endothelial cells in patients with severe disease would be more permissive to virus replication than in mild cases, and lead to greater monocyte activation. Rather, other factors including host genetics may play a role in the propensity of monocytes to upregulate CCR7 in response to infected endothelial cells, although this is currently only our speculation. All monocytes displayed increased adhesion to PUUV-infected and to TNFα-treated endothelial cells suggesting a general increase in the adhesive properties of endothelial cells towards monocytes during acute HFRS. This could lead to a transient loss of monocytes from circulation, which are actively replenished by monocyte mobilization from the bone marrow. However, to understand the mechanism by which monocytes adhere to an infected endothelium in the context of PUUV infection, and whether they contribute to local inflammation and/or vascular leakage upon infiltration to the kidneys, further systematic in vitro studies are required.

Taken together our results suggest that during severe PUUV-caused HFRS, monocytes are activated upon exposure to PUUV in blood, stimulating endothelial adhesion and subsequent redistribution from circulation to the kidneys. Our data indicate that monocytes contribute to HFRS pathogenesis and disease severity during acute PUUV infection.

## Material and methods

### Ethics statement and patient material

The study was approved by the Ethics Committee of Tampere University Hospital (Nos. 99256, R18808 and R04180). All subjects gave written informed consent in accordance with the Declaration of Helsinki. The study consisted of peripheral blood mononuclear cells (PBMC) and plasma samples from 23 patients treated for serologically confirmed acute PUUV infection at the Tampere University Hospital, Finland, during 2002–2007. The samples were collected on sequential days during hospitalization (5–10 days after symptom onset) and 1–3 times during recovery on 11–30, 31–50, 180 or 360 days after symptom onset. Prior to analysis PBMC and plasma samples were stored in liquid nitrogen and -80˚C, respectively. Patients were stratified as having either severe or mild HFRS by adapting the SOFA (sequential organ failure assessment)-based scoring system where the maximum levels of plasma creatinine (4 = > 440, 3 = 300–440, 2 = 171–299, 1 = 110–170 and 0 = < 110 μmol/l), minimum level of thrombocytes (4 = < 20, 3 = 20–49, 2 = 50–99, 1 = 100–150 and 0 = > 150 x $10^9$/L) and a

lowest mean arterial blood pressure measured during hospitalization (1 = < 70 and 0 = ≥ 70 mmHg) were ranked. A total adapted-SOFA (a-SOFA) score of ≥ 5 (12 patients) was considered as severe and < 5 (11 patients) as mild. Similar adapted scores have been used for HFRS previously [53]. A set of 80 individual archival (from years 1978–1988) bouin-fixed paraffin-embedded kidney biopsy specimens from patients with acute HFRS (n = 37, 7–17 days after symptom onset) or other kidney diseases (n = 41) were also obtained (listed in S2 Table). The Finnish national supervisory authority of health and welfare approved the study on kidney biopsies (No. V/19454/2018)

## Sample collection

Venous blood samples were collected from all patients (acute and convalescent phase samples) and uninfected controls in the study in 6 vacutainers containing EDTA as an anticoagulant and stored on ice. Blood samples were centrifuged at 800g for10 min in room temperature (RT) and plasma was collected and frozen at -70˚C. The blood volume was reconstituted with RPMI (HyClone). Histopaque was added to Accuspin tubes and centrifuged at 800g for 30sec at RT. PBMCs were obtained by density-gradient centrifugation after centrifugation at 800g for 20 min at RT (without brake and cryopreserved in RPMI + 20% inactivated fetal calf serum (FCS) (Gibco) + 10% DMSO (Sigma) and stored in liquid nitrogen.

## Flow cytometry

Phenotypic analysis was performed on PBMCs using panels containing Live/Dead Blue (Life technologies), CD1c (AD5-8E7; Miltenyi), CD3 (SP34-2 and SK7; BD), CD10 (HI10a; Biolegend), CD11c (B-Ly6, BD), CD14 (M5E2, BD), CD16 (3GE, Biolegend), CD19 (HIB19; Biolegend), CD20 (L27, BD), CD34 (561, Biolegend), CD45RA (HI100, Biolegend), CD56 (HCD56, BD), CD62L (SK11, BD), CD66abce (TET2, Miltenyi Biotec), CD80 (L307.4, BD), CD86 (2331; BD); CD123 (7G3; BD), CD133 (7, Biolegend), CD141 (AD5-14H12; Miltenyi), CD146 (P1H12; Biolegend), HLA-DR (TU36, Life technologies), CCR2 (K036C2, Biolegend), CCR4 (L291H4, Biolegend), CCR6 (11A9, BD) and CCR7 (150503, BD) (Panels summarized in S3 Table). Fixation was performed with 1% paraformaldehyde. Samples were acquired on an LSRFortessa flow cytometer (BD Biosciences) and analyzed using FlowJo software (Tree Star).

## Stimulation with TLR ligands and intracellular cytokine analysis

PBMCs were stimulated with 1 ng/mL 3M-019 (TLR 7/8 agonist) (Invivogen) for 3 hours (h) in R10 [RPMI-1640 (Sigma-Aldrich) medium supplemented with 10% FCS, 5mM L-glutamine, 100U/mL each of penicillin and streptomycin (all Invitrogen)] and incubated at 37˚C/5% CO2. Brefeldin A (Sigma-Aldrich) was added at 10 mg/mL, 30 minutes after TLR stimulation. Cells were stained with surface antibodies as above, followed by fixation and permeabilization using a staining buffer set (eBioscience). Intracellular cytokine staining was performed using TNFα (Mab11, BD) & IL-6 (MQ2-13A5, Biolegend) antibodies. Samples were acquired on an LSRFortessa flow cytometer (BD Biosciences) and analyzed using FlowJo software (Tree Star).

## Cytokine and chemokine assays

Cytokines and chemokines in plasma (1:2 dilution) were measured using a custom-designed 12-plex Luminex assay (for TNFα, IL-6, IFNα, IFNγ, IL-8, IL-10, IL-18, CCL2, CCL3, CCL7, IL-1β and IL-12p70) kit (R&D Systems) and analyzed on the Bio-Plex 200 instrument (Bio-Rad). Exact values were interpolated from standard curves (5PL logistic regression) on built-in

Bio-Plex Software (Bio-Rad). M-CSF and GM-CSF in plasma was measured by enzyme-linked immunosorbent assay (ELISA) from Sigma and R&D Systems, respectively, according to manufacturer's instructions. Exact values were interpolated from standard curves (5PL logistic regression) on GraphPad Prism version 9.0 (GraphPad Software).

## Viruses

The PUUV-Suonenjoki (PUUV-Suo) strain was propagated in a bank vole renal epithelial cell line (MyGlaRec.B from EVAg), grown in Dulbecco's minimum essential medium-high glucose (DMEM; Sigma Aldrich) supplemented with 10% inactivated FCS, 100 IU/ml Penicillin, 100 μg/ml Streptomycin, 2mM L-glutamine and a mix of non-essential amino acids (Sigma Aldrich). Virus was purified from cell culture supernatants by ultracentrifugation through 30% sucrose cushion (SW28 rotor, 27,000 rpm, 50 min, +4˚C) and suspended to the corresponding growth medium. The infectious titer of PUUV stocks were routinely ~$10^6$–$10^7$ focus-forming units (FFU)/ml. Where indicated, PUUV was inactivated using UV crosslinker (300,000 μJ/cm$^2$, Stratalinker, Stratagene). For mock infections, conditioned cell culture supernatant without virus infection was prepared in the same fashion as for virus preparations.

## In vitro monocyte activation assays

Total monocytes were enriched from buffy coats of donors obtained from the Finnish Red Cross blood service with RosetteSep monocyte enrichment kit (STEMCELL Technologies). After washing, CD16$^+$ monocytes were isolated using CD16 magnetic beads (Miltenyi Biotech) and the remaining CD16– cells (confirmed to primarily consist of CD14+ monocytes) were used in parallel. For activation experiments, monocytes were suspended to 1 million cells per ml R10 medium. For direct virus exposure experiments, 250,000 monocytes were incubated with PUUV at multiplicity of infection (MOI) 1, or equal volumes of UV-inactivated PUUV or mock preparations for 1 h at 37˚C. Cells were centrifuged (400g for 10 min at 4˚C), virus inoculum removed and suspended back to 1 million cells/ml in R10. 1 ng/mL 3M-019 (TLR 7/8L) (Invivogen) was used as a positive control. After incubation for 19 hr at 37˚C, cells were stained for multicolor flow cytometry as described above. To study monocyte interactions with PUUV-infected endothelial cells, human primary dermal blood microvascular endothelial cells (BECs), grown in endothelial basal medium (EBM-2) supplemented with SingleQuots Kit (all from Lonza) and used at passages 7–10, were infected with PUUV at MOI of 10 or equal volumes of UV-inactivated PUUV for 1 h at 37˚C. For negative and positive controls, BECs were either left untreated or treated with 50 ng/ml TNFα (R&D systems), respectively. 72h post infection, cell supernatants were removed and cells were washed. Isolated naïve monocytes were added to PUUV-infected or TNFα-treated BEC monolayers in a 2.5:1 cell ratio (250 000 monocytes / 100,000 BECs), incubated for 20 hr at 37˚C, cells scraped with rubber policeman, washed, strained and analysed by multicolor flow cytometry as described above.

## Confocal imaging

To visualize monocyte adhesion to PUUV-infected endothelial cells, BECs were infected or TNFα-treated with PUUV as described above, in 96-well plates (ViewPlate-96 black, PerkinElmer), at a density of 10,000 cells per well. After 72 h isolated naïve monocytes were added to BEC monolayers in a 5:1 cell ratio (100 000 monocytes / 20 000 BECs) and incubated for 1 h at 37˚C. Cells were washed twice with PBS, fixed with 4% paraformaldehyde for 10min and blocked/ permeabilized (3% BSA; 0.1% Triton-X100 in PBS) for 10min. Cells were incubated with mouse monoclonal anti-human HLA-DR antibody (HI43, Immunotools) followed by

secondary AlexaFluor488-conjugated anti-mouse secondary antibody (Thermo Scientific) both for 1hr at room temperature. Cells were washed three times with PBS before imaging with PerkinElmer Opera Phenix spinning disk confocal microscope using a 20x water-immersion objective (NA 1.0). The analysis of the positive monocytes was conducted with the Harmony software (PerkinElmer) by using a supervised linear classifier. The analysis pipeline and intensity, morphology and texture features used in the classification are listed in S1 Table.

## Immunohistochemistry and quantification

Paraffin-embedded kidney biopsy specimens from acute HFRS and unrelated kidney diseases as controls were stained with HLA-DR, CD14, CD16 and CD68 antibodies followed by polymer-based horseradish peroxidase (HRP) chromogenic detection and hematoxylin counterstain. All stainings were performed based on standard procedures at Helsinki University diagnostic laboratory HUSLAB. Digital images were obtained with Zeiss Axioscan slide scanner and quantification of HLA-DR, CD14, CD16 and CD68 positive cells as compared to total number of cells was performed using Aiforia v4.8 cloud platform (Aiforia Technologies Oy, Helsinki, Finland). This platform uses image-based artificial intelligence (AI) with high-performance cloud computing to discriminate between HRP-positive and -negative cells in individual tissues. The AI training process included ~2500 iterations based on ~500 annotations of manually selected positive and negative objects with visible nuclei.

## Statistics

Data were analyzed using SPSS software v 25.0 (IBM Corp.), GraphPad Prism version 6.0 (GraphPad Software) and JMP, version 14.2. (SAS Institute Inc., Cary, NC, 1989–2019). Cell frequencies and cell surface marker expression levels at various time points during acute and convalescent stage HFRS were assessed by generalized estimating equations (GEE) assuming gamma distribution with log link function for the outcome variable of interest and setting the last convalescence time point (360 days) as the reference category (SPSS). Sequential Sidak's test was used to control for multiple comparisons. The predictive value of measured variables in determining severe or mild HFRS was also assessed by GEE. In this case the model included the main effects of time points and their interaction with the variable of interest. Only the earliest individual patient sample at each time point was included for GEE analysis. The working correlation matrix for GEE analysis was set to unstructured. For comparisons in TLR-stimulation experiments, mixed-effects analysis was performed using Dunnett's multiple correction test. Differences between groups in *in vitro* experiments were assessed using one-way ANOVA with Kruskal-Wallis test, and Dunn's test to correct for multiple comparisons. Bivariate and multivariate linear regression analysis was performed using JMP, choosing Spearman's rank correlation coefficient for nonparametric analyses. Data were considered significant at $p < 0.05$.

## Supporting information

**S1 Fig. CD34+ progenitors are increased in acute HFRS patients.** (**A**) Gating strategy for identification of CD34+ progenitors from PBMCs by flow cytometry. (**B**) Flow plots depict CD34+ progenitors in a representative uninfected control and an HFRS patient over time (day 9-day 360). (**C**) Graph summarizes the frequencies of CD34+ cells in UCs (n = 3) and HFRS patients (n = 16). Statistical differences between day 180 (too few data points at day 360) and other time points were assessed using a generalized estimated equation (GEE) model in SPSS and differences were considered significant at $p < 0.05$ (**$p < 0.01$ and ****$p < 0.0001$). (TIF)

**S2 Fig. Gating strategy used for identification of monocytes and dendritic cells from PBMCs.** (**A**) From total cells, live cells were identified as single cells negative for the live/dead marker. The lineage negative, HLA-DR+ cells were identified as cells expressing HLA-DR, but not CD3, CD19, CD20, CD56 or CD66. Myeloid cells were identified as CD11c-expressing cells and from the CD11c– population, CD123+ cells were identified as plasmacytoid dendritic cells (PDCs). From the CD11c+ cells, monocyte subsets were identified as CD14+CD16– classical monocytes (CM), CD14+CD16+ intermediate monocytes (IM) and CD14–CD16+ nonclassical monocytes (NCM). Myeloid DC subsets (CD1c+ and CD141+ MDCs) were identified from the CD14–CD16– cells. (**B**) Histograms show the relative expression of HLA-DR, CCR2, CD62L and CCR7 in classical, intermediate and nonclassical monocytes of a single representative HFRS patient over time (day 8 to day 360).
(TIF)

**S3 Fig. Depletion of myeloid dendritic cells in blood from patients during HFRS as compared to uninfected controls.** (**A**) Gating strategy for identification of dendritic cells (DCs) from PBMCs by flow cytometry. Representative uninfected control (UC) sample showing gating on CD11c+ myeloid cells which were negative for CD14 and CD16 to identify CD1c+ (coral) and CD141+ (maroon) myeloid DC subsets. From the CD11c– cells, CD123+ plasmacytoid DCs (teal) were identified. (**B**) Plots depict the myeloid DC (MDC) populations in a UC and a representative HFRS patient over the course of disease (day 6–360). (**C-E**) Graphs show frequencies (to total live cells) of (**C**) CD141+ MDCs, (**D**) CD1c+ MDCs and (**E**) PDCs in PBMCs from patients (filled bars) and HCs (empty bars). Statistical differences between day 360 and other time points were assessed using a generalized estimated equation (GEE) model in SPSS and differences were considered significant at $p < 0.05$. (**F-H**) Graphs show median ± IQR frequencies of (**F**) CD141+ MDCs, (**G**) CD1c+ MDCs and (**H**) PDCs in PBMCs from patients stratified by severity as lower a-SOFA score (<5, empty circles) or higher a-SOFA score (≥5, filled circles). Median frequencies of respective DC subset in UCs are indicated with dotted lines.
(TIF)

**S4 Fig. Gating strategy used for identifying TNFα and IL-6 producing myeloid cells with or without TLR stimulation.** (**A**) Flow plots depict **TNFα**-producing CD11c+ myeloid cells (gated as shown in S1 Fig) in a representative uninfected control and an HFRS patient over time (day 8-day 360), in the absence (top) or presence of TLR 7/8L stimulation (bottom) in vitro for 3hr. (**B**) Flow plots depict IL-6-producing CD11c+ myeloid cells (gated as shown in S1 Fig) in a representative uninfected control (UC) and an HFRS patient over time (day 8-day 360), in the absence of TLR 7/8L stimulation (bottom) in vitro for 3hr, and the frequencies are summarized in (**C**). (**D**) Graph displays frequency of IL-6 producing cells in CD11c+ myeloid cells in PBMCs in UCs (n = 3) and HFRS patients (n = 15) in the absence (circle) or presence (square) of TLR 7/8L stimulation for 3 hr. Statistical differences between groups (of similar exposure conditions) were assessed by mixed-effects analysis using Dunnett's multiple correction test and considered significant at $p < 0.05$. (**E**) Lines display bivariate linear regression analysis between thrombocyte counts ($10^9$/L) and frequency of IL-6 producing CD11c+ myeloid cell without (left) or with (right) TLR 7/8L stimulation. The shaded area represents the 95% confidence region for the fitted line. Patients are stratified by severity as lower, i.e. a-SOFA score <5 (blue) or higher, i.e. a-SOFA score ≥5 (pink). ρ represents Spearman rand statistical differences were considered significant at $p < 0.05$.
(TIF)

**S1 Table. Image analysis pipeline parameters for Harmony software.**
(XLSX)

**S2 Table. Clinical characteristics of patients with kidney biopsies.**
(XLSX)

**S3 Table. Flow cytometry panels.**
(XLSX)

## Acknowledgments

We thank the patients and volunteers who have contributed clinical material to this study. We would like to thank Sanna Mäki, Irina Suomalainen at University of Helsinki, as well as Kati Ylinikkilä and Eini Eskola at Tampere University for technical assistance. The authors would like to thank FIMM high-content imaging and analysis unit (FIMM-HCA) and Biocenter Finland for technical services and assistance.

## Author Contributions

**Conceptualization:** Sindhu Vangeti, Tomas Strandin, Sang Liu, Jukka Mustonen, Antti Vaheri, Olli Vapalahti, Jonas Klingström, Anna Smed-Sörensen.

**Data curation:** Sindhu Vangeti, Tomas Strandin, Sang Liu, Johanna Tauriainen, Anne Räisänen-Sokolowski, Antti Hassinen, Satu Mäkelä, Jukka Mustonen, Antti Vaheri, Anna Smed-Sörensen.

**Formal analysis:** Sindhu Vangeti, Tomas Strandin, Sang Liu, Johanna Tauriainen, Anne Räisänen-Sokolowski, Luz Cabrera, Satu Mäkelä, Antti Vaheri, Olli Vapalahti, Anna Smed-Sörensen.

**Funding acquisition:** Sindhu Vangeti, Tomas Strandin, Jukka Mustonen, Antti Vaheri, Olli Vapalahti, Jonas Klingström, Anna Smed-Sörensen.

**Investigation:** Sindhu Vangeti, Tomas Strandin, Sang Liu, Johanna Tauriainen, Anne Räisänen-Sokolowski, Luz Cabrera, Satu Mäkelä, Jukka Mustonen, Antti Vaheri, Olli Vapalahti, Jonas Klingström.

**Methodology:** Sindhu Vangeti, Tomas Strandin, Sang Liu, Anne Räisänen-Sokolowski, Luz Cabrera, Jukka Mustonen, Jonas Klingström, Anna Smed-Sörensen.

**Project administration:** Sindhu Vangeti, Tomas Strandin, Jukka Mustonen, Antti Vaheri, Olli Vapalahti, Anna Smed-Sörensen.

**Resources:** Tomas Strandin, Anne Räisänen-Sokolowski, Antti Hassinen, Satu Mäkelä, Jukka Mustonen, Antti Vaheri, Olli Vapalahti, Jonas Klingström, Anna Smed-Sörensen.

**Software:** Tomas Strandin, Antti Hassinen, Jonas Klingström.

**Supervision:** Tomas Strandin, Jukka Mustonen, Antti Vaheri, Olli Vapalahti, Anna Smed-Sörensen.

**Validation:** Sindhu Vangeti, Tomas Strandin, Sang Liu, Anne Räisänen-Sokolowski, Antti Hassinen, Satu Mäkelä, Jukka Mustonen, Olli Vapalahti, Anna Smed-Sörensen.

**Visualization:** Sindhu Vangeti, Tomas Strandin, Anne Räisänen-Sokolowski, Antti Hassinen.

**Writing – original draft:** Sindhu Vangeti, Tomas Strandin.

**Writing – review & editing:** Sindhu Vangeti, Tomas Strandin, Sang Liu, Johanna Tauriainen, Anne Räisänen-Sokolowski, Luz Cabrera, Antti Hassinen, Satu Mäkelä, Jukka Mustonen, Antti Vaheri, Olli Vapalahti, Jonas Klingström, Anna Smed-Sörensen.

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
