## [Decision Letter · Decision Letter 0]

10 Nov 2020

Dear Dr. Anna Smed-Sörensen,

Thank you very much for submitting your manuscript "Monocyte subset redistribution from blood to kidneys in patients with Puumala virus caused hemorrhagic fever with renal syndrome" for consideration at PLOS Pathogens. As with all papers reviewed by the journal, your manuscript was reviewed by members of the editorial board and by several independent reviewers. In light of the reviews (below this email), we would like to invite the resubmission of a significantly-revised version that takes into account the reviewers' comments.

We cannot make any decision about publication until we have seen the revised manuscript and your response to the reviewers' comments. Your revised manuscript is also likely to be sent to reviewers for further evaluation.

Sincerely,

Christina F Spiropoulou

Guest Editor

PLOS Pathogens

Adolfo García-Sastre

Section Editor

PLOS Pathogens

Kasturi Haldar

Editor-in-Chief

PLOS Pathogens

orcid.org/0000-0001-5065-158X

Michael Malim

Editor-in-Chief

PLOS Pathogens

orcid.org/0000-0002-7699-2064

Reviewer's Responses to Questions

**Part I - Summary**

Reviewer #1: In their manuscript entitled “Monocyte subset redistribution from blood to kidneys in patients with Puumala virus caused hemorrhagic fever with renal syndrome” Vangeti et al. analysed redistribution of monocytes in longitudinal peripheral blood samples from patients infected with Puumala virus (PUUV), which causes a mild form of hemorrhagic fever with renal syndrome (HFRS). They found that in peripheral blood of acute phase HFRS patients nonclassical monocytes (NCMs, CD14+CD16++) are depleted whereas classical (CMs, CD14++ CD16−) and intermediate monocytes (IMs, CD14++CD16+) are increased. These data are in contrast with a previous publication of the authors (Scholz et al., 2017), in which they described diminished numbers of all monocyte subsets (CM, IM, and NCM) in the peripheral blood of patients with acute HFRS. As reported previously, the expression of activation markers such as CCR7 on blood lymphocytes suggests that monocytes are activated and migrate to peripheral tissue during acute HFRS. In vitro, monocytes upregulated CCR7 after direct exposure to PUUV (CD16+ monocytes) or after contact with PUUV-infected endothelial cells (CD16- monocytes) supporting this assumption. Previously, the authors described a significant influx of monocytes expressing HLA-DR, CD11c or CD123 in bronchial tissue of patients with PUUV infection (Scholz et al., 2017). In the present manuscript, increased numbers of HLA-DR+, CD14+ and CD16+ cells were detected in the renal tissues of PUUV-infected patients. Based on their findings, the authors suggest that NCMs are recruited to the kidneys during HFRS. The analysis of kidney biopsies is interesting but incomplete. There are several problematic issues as described below.

Reviewer #2: The manuscript by Vangeti et al. describes experiments in which they examined the different monocyte subsets in PUUV infected patients. In the study, longitudinal peripheral blood samples and renal biopsies were examined from HFRS patients infected by PUUV and in vitro experiments were performed. This type of study can provide insights into the immunopathogenesis of hemorrhagic fever with renal syndrome caused by PUUV. The manuscript is concise and well written. In my opinion, the manuscript provide some answers of what is the role of monocytes in PUUV caused HFRS. Immunopathogenesis is an important area of research in the hantavirus field. These results are important to share and merit publication, however I have some minor comments and suggestions.

Reviewer #3: The authors report declines of CD16+ monocytes in the blood during the acute phase of HFRS illness and this correlated with increased CCR2 expression in this cell population c/w the hypothesis that these cells could be getting recruited to sites of inflammation from the blood stream. Of particular interest was the inclusion of pathologic specimens from renal biopsies of HFRS patients showing increased HLADR, CD14 and CD16 as compared to other renal pathologies. The reported vitro experiments are of limited value to this study given the difficultly of interpreting their relevance to in vivo conditions. Nevertheless, this is a well written and important study of monocytes in patients with HFRS. These are difficult to perform studies given the rarity of this disease and the challenges associated with human subjects research.

**Part II – Major Issues: Key Experiments Required for Acceptance**

Reviewer #1: Point 1:

In Figure 2D the differences between patients a-SOFA score < 5 and patients with a-SOFA score ≥ 5 are not significant. In addition, the interquartile range (IQR) shows a high variation. Moreover, it is unclear in how many samples and in how many patients the percentage of CD34+ cells have been determined. From my point of view, Figure 2D could be removed. Along this line, the authors should explain, why an increased frequency of CD34+ hematopoietic progenitors in peripheral blood suggests that monocytes are mobilized from the bone marrow during acute HFRS (see Discussion, line 304-306).

Point 2:

Figure 4 (showing kidney biopsies) and Figure 5 (showing cytofluorimetric analyses) are mixed up. According to the figure legends, Figure 4 should be labelled Figure 5 and vice versa.

Point 3:

In Figure 5 (labelled Figure 4) it is important to know the kidney diseases of the controls and other clinical data of the controls especially age distribution

Point 4:

The authors claim that monocytes are found in increased numbers in the kidneys of patients with acute HFRS. It is unclear, however, which cell type is really observed in the kidney biopsies from HFRS patients. Under inflammatory conditions, monocytes that extravasate into peripheral tissues develop into another cell type with morphological and functional features of dendritic cells (DC) or macrophages. For this reason, the HLA-DR+ and CD14+ or CD16+ cells detected in the kidney biopsies are not monocytes but monocyte-derived inflammatory DCs or macrophages. The authors should analyse additional markers to clarify this important point.

Point 5:

The extravasation of monocytes into kidney tissue implies that kidney tissue is infected by PUUV. Indeed, a previous study of PUUV-infected patients has shown that hantaviral N protein is detected in podocytes in the glomeruli and in tubular epithelial cells (Krautkrämer et al., 2011). However, in the present study the monocyte-derived inflammatory cells localized predominantly to the tubulointerstitial space. It would be important to determine whether hantaviral N protein can be detected and whether it is located in the proximity of infected areas. Moreover, the authors should discuss how the virus is transported from the lung to the kidney.

Point 6:

In order to determine whether HFRS directly impacted on blood monocyte function the author cultured and/or stimulated blood monocytes from HFRS patients and uninfected controls in vitro for 3 hours. They observed downregulation of CD16 indicating that the cells are no longer monocytes but developed into inflammatory DCs or macrophages. It has been shown that monocytes stimulated with hantavirus differentiate into DC-like cells (Markotic et al., 2007; Raftery et al., 2020). Thus, the authors should determine in which cell type monocytes develop under the conditions used.

Point 7:

The rationale behind stimulating monocytes with agonists of TLR7/8 is unclear, as hantaviruses are not known to trigger TLR7/8. In contrast, TLR3 and RIG-I-like receptors have been shown to function as PRRs for hantavirus infection (Handke et al., 2009; Lee et al., 2011; Kell et al., 2020). Thus, the observed impairment of TLR7/8 signaling pathways in monocytes from HFRS patients (Figure 6C) could be irrelevant for resolution of hantavirus infection.

Point 8:

In Figure 6A, the differences between IL-6 production of stimulated versus unstimulated myeloid cells do not seem to be significant and the interquartile range (IQR) shows a high variation. In addition, the difference between IL-6 production of UCs versus HFRS patients does not seem to be significant.

Point 9:

In a previous study of PUUV-infected patients in Northern Sweden, the authors reported diminished numbers of all monocyte subsets (CM, IM, and NCM) and DCs subsets in peripheral blood during the acute stage of HFRS (Scholz et al., 2017). This is in stark contrast to the present study, in which acutely PUUV-infected patients from Central Finland showed increased numbers of circulating CMs and IMs whereas the numbers of NCMs were decreased. The authors explain this striking difference with an overall higher severity of HFRS in Central Finland. This explanation is not convincing as the Finish patients with a-SOFA < 5 (less severe disease) also show the same pattern: increased numbers of circulating CMs and IMs and decreased numbers of NCMs.

Point 10:

In Figure 7B-G, the differences between patients with a-SOFA score < 5 and patients with a-SOFA score ≥ 5 are not significant. Moreover, there is confusion regarding Figure 7G. In the figure legend, Figure 7G (IL-18 and CCL2; against thrombocyte counts (109/L) is mentioned but not shown in the Figure. In the text, Figure 7G is referred to as showing elevated plasma creatinine levels correlating with thrombocytopenia.

Point 11:

The headline of the last paragraph “CD16+ but not CD16– monocytes upregulate CD62L in response to direct virus exposure in vitro” is contradictory to the statement later in the paragraph (line 232-233) claiming that “Both CD16+ and CD16– monocytes upregulated CD62L after direct exposure to PUUV…”.

Point 12:

In Figure 8B, it is important to know how many monocytes are infected with PUUV after direct exposure as compared to monocytes exposed to virus-infected endothelial cells. It cannot be ruled out that monocytes exposed to virus-infected endothelial cells are less efficiently infected and, therefore, do not upregulate CD62L. Moreover, differential susceptibility to PUUV infection could explain the differences of surface expression between CD16- and CD16+ monocytes.

Reviewer #2: (No Response)

Reviewer #3: 1. A demonstration of the levels of GMCSF and MCSF in the patient plasma would significantly strength the authors hypothesis that loss of CD16+ monocytes from the blood leads to BM mobilization of progenitors. These are readily available luminex assays and levels of one or both would be expected to be elevated in the acute phase, and likely also higher in the more severely affected patients.

**Part III – Minor Issues: Editorial and Data Presentation Modifications**

Reviewer #1: Point 1:

For sake of clarity, in the legends to Figure 6 and Figure 8 the authors should describe the shown experiments in more detail. For example, it is unclear how long the mononuclear cells were cultured/stimulated in vitro and the MOI used for infection is not mentioned.

Point 2:

In Figure 4B, the authors show that CCR2 expression on CM is not affected during acute HFRS. In their previous paper (Scholz et al., 2017), however, the authors report that CCR2 was downregulated on classical monocytes (CM) after PUUV infection in vitro. The authors should comment on this difference.

Reviewer #2: I have some comments about Figures:

1. There is a substitution between Figure 4 and Figure 5. Figure 5 was uploaded instead of Figure 4 and Figure 4 instead of Figure 5.

2. On Figure 7 is correct 7G instead of 7H.

3. Figure 8D is not mentioned in the text (at results section).

Line 337: It is known or suspected which host genetics factors have a role in the monocyte response to infected endothelial cells?

Line 403: I would include more detailed information about performance of Luminex assay.

Reviewer #3: Line 48: "noticeable" burden- as compared to an unnoticeable burden?

Table 2: units for Creatinine are incorrect.

There is no description of the disease course for the patients whose renal biopsies are used. Acknowledging that these samples are from 40+ years ago, can some clinical data be obtained? Day post symptom onset that the samples were obtained? Cre levels? Demographics?

T cell staining in the renal biopsies is also an important addition, since activated T cells could also be HLADR+ and the authors prior data demonstrated increased T cells in the lungs of HFRS patients.

Authors should be careful to not attribute monocyte infiltration into the kidney to the pathology. Only correlation is demonstrated in this study. There are multiple instances of this throughout the manuscript.

Line 225: title to the section does not represent the data presented, since CD16- monocytes do up-regulate CD62L, just to a lesser extent. From the paragraph: "Both CD16+ and CD16– monocytes upregulated CD62L after direct exposure to PUUV or TLR7/8L, but in both cases CD16+ monocytes displayed higher level of CD62L induction than CD16– monocytes (Figure 8B, top panel)."

Line 247: Be careful to not attribute a clinical patient finding based upon an in vitro expt.

For flow: make sure representative gating strategy is shown for all analyses. There isn't one for CCR2/7 staining or for IL-6, TNFa in vitro expts. Also, need to make a table showing exact panel composition for each panel including the fluorphores used- can be supplementary.

Figure 6: Please show all data points for this figure rather than bar graphs so reader can appreciate the spread of the data.

For in vitro expts different ratios of monocytes to endothelial cells were used depending upon the expt. How were these ratios determined? How do they compare with what would be expected in vivo?

PLOS authors have the option to publish the peer review history of their article (what does this mean?). If published, this will include your full peer review and any attached files.

Reviewer #1: No

Reviewer #2: No

Reviewer #3: No
---

## [Decision Letter · Decision Letter 1]

17 Feb 2021

Dear Anna Smed-Sorensen

We are pleased to inform you that your manuscript 'Monocyte subset redistribution from blood to kidneys in patients with Puumala virus caused hemorrhagic fever with renal syndrome' has been provisionally accepted for publication in PLOS Pathogens.

Best regards,

Christina F Spiropoulou

Guest Editor

PLOS Pathogens

Adolfo García-Sastre

Section Editor

PLOS Pathogens

Kasturi Haldar

Editor-in-Chief

PLOS Pathogens

orcid.org/0000-0001-5065-158X

Michael Malim

Editor-in-Chief

PLOS Pathogens

orcid.org/0000-0002-7699-2064

Reviewer Comments (if any, and for reference):

Reviewer's Responses to Questions

**Part I - Summary**

Reviewer #1: The questions were answered comprehensively. Many thanks

Reviewer #3: My prior comments have been addressed.

**Part II – Major Issues: Key Experiments Required for Acceptance**

Reviewer #1: (No Response)

Reviewer #3: (No Response)

**Part III – Minor Issues: Editorial and Data Presentation Modifications**

Reviewer #1: (No Response)

Reviewer #3: (No Response)

PLOS authors have the option to publish the peer review history of their article (what does this mean?). If published, this will include your full peer review and any attached files.

Reviewer #1: No

Reviewer #3: No

---

## [Editor Report · Acceptance letter]

7 Mar 2021

Dear Dr. Smed-Sörensen,

We are delighted to inform you that your manuscript, "Monocyte subset redistribution from blood to kidneys in patients with Puumala virus caused hemorrhagic fever with renal syndrome," has been formally accepted for publication in PLOS Pathogens.

Best regards,

Kasturi Haldar

Editor-in-Chief

PLOS Pathogens

orcid.org/0000-0001-5065-158X

Michael Malim

Editor-in-Chief

PLOS Pathogens

orcid.org/0000-0002-7699-2064